# Thymic Carcinoma: Unraveling Neuroendocrine Differentiation and Epithelial Cell Identity Loss

**DOI:** 10.3390/cancers16010115

**Published:** 2023-12-25

**Authors:** Yosuke Yamada, Kosuke Iwane, Yuki Nakanishi, Hironori Haga

**Affiliations:** 1Department of Diagnostic Pathology, Kyoto University Hospital, Kyoto 606-8507, Japan; haga@kuhp.kyoto-u.ac.jp; 2Department of Gastroenterology and Hepatology, Kyoto University Graduate School of Medicine, Kyoto 606-8507, Japan; iwanekosuke@kuhp.kyoto-u.ac.jp (K.I.); yuki@kuhp.kyoto-u.ac.jp (Y.N.)

**Keywords:** thymic epithelial tumors, thymoma, thymic carcinoma, tuft cells, ionocytes, neuroendocrine cells, HDAC9, NFATC1, PAX9, SIX1

## Abstract

**Simple Summary:**

Comprehensive genomic profiling has significantly enhanced the molecular understanding of thymic epithelial tumors (TETs). However, due to the general absence of druggable mutations in TETs, their expression signatures could provide an alternative avenue for identifying new drug targets. In the present study, we broadly examined the neuroendocrine phenotypes of thymic carcinoma and identified upregulated genes potentially associated with these phenotypes. Our findings revealed characteristic expressions of HDAC9 and NFATC1—which are both potential therapeutic targets—in the neuroendocrine group of adult thymic epithelial cells. These genes were also significantly more expressed in thymic carcinomas than in thymomas. Additionally, we discovered that pan-thymic epithelium markers, exemplified by *PAX9* and *SIX1*, were notably suppressed in thymic carcinomas, suggesting another distinctive feature of this tumor type.

**Abstract:**

Background: The histogenesis of thymic epithelial tumors (TETs) has been a subject of debate. Recent technological advancements have revealed that thymic carcinomas often exhibit a phenotype akin to tuft cells, which is a subset of medullary TECs. Here, we further explored the gene expression signatures of thymic carcinomas in relation to tuft cells and their kinships—ionocytes and neuroendocrine cells (neuroendocrine group). Methods: We analyzed a single-cell RNA sequencing dataset from the normal human thymus. Concurrently, we examined publicly available datasets on the mRNA expression and methylation status of TECs and lung cancers. Real-time quantitative PCR was also conducted with our tissue samples. Results: Thymic carcinomas displayed a neuroendocrine phenotype biased toward tuft cells and ionocytes. When exploring the possible regulators of this phenotype, we discovered that *HDAC9* and *NFATC1* were characteristically expressed in the neuroendocrine group in adult TECs and thymic carcinomas. Additionally, the pan-thymic epithelium markers, exemplified by *PAX9* and *SIX1*, were significantly suppressed in thymic carcinomas. Conclusions: Thymic carcinomas might be characterized by unique neuroendocrine differentiation and loss of identity as thymic epithelial cells. Future studies investigating the role of HDAC9 and NFATC1 in thymic epithelium are warranted to explore their potential as therapeutic targets in TETs.

## 1. Introduction

The histogenesis of thymic epithelial tumors (TETs) has been the subject of extensive research, with insights drawn from normal thymic physiology and the technological tools available at the time [1,2,3,4,5,6,7,8,9]. Immunohistochemical studies have played a pivotal role in this exploration, leading to the widely accepted notion that type B thymomas, particularly the B2 subtype [10], largely exhibit a phenotype of cortical thymic epithelial cells (cTECs).

However, the histogenesis of thymic carcinoma, in which squamous cell carcinoma (SQCC) accounts for 80% of the cases, has remained inconclusive. Recent advancements in technology, including single-cell RNA sequencing and the resulting publicly available datasets, have shed light on the detailed expression profiles of normal TECs and TETs. These tools have enabled us to investigate the properties of TETs in relation to the TEC properties. Following the discovery of heterogeneity in mouse medullary TECs (mTECs), which can be divided into four cell types, including the tuft cell subset [11,12,13,14,15], and comprehensive molecular profiling of TETs by the Cancer Genome Atlas (TCGA) [16,17,18], our group found that thymic carcinomas often exhibited a tuft cell-like gene/protein expression signature [19,20,21].

The TEC diversity was later confirmed in humans [22]. According to this study, human cTECs and mTECs can be divided into two and eight cell types, respectively. The latter includes not only tuft cells but also ionocytes [23,24] and neuroendocrine cells as a rare epithelial subset [22]. Using this dataset, Matsumoto et al. proposed that thymic carcinomas have an mTEC nature, exemplified by the AIRE expression [7]. They also found that common diagnostic markers of thymic SQCC were characteristically expressed in tuft cells/ionocytes and neuroendocrine cells, with *KIT* and *MUC1* expressed in the tuft cells/ionocytes and *CD5* in the neuroendocrine cells [25,26,27,28]. Regarding KIT, this result was consistent with the findings of previous studies that demonstrated a strong correlation between tuft cell lineage and KIT expression [19,29]. Considering the developmental relationship among these three cell types in the lungs [24] and that tuft cell-like cancers were first discovered as a subset of small cell lung carcinoma (SCLC) [29], a representative neuroendocrine carcinoma, tuft cells, and ionocytes might be broadly regarded as neuroendocrine cells. Furthermore, general neuroendocrine markers, including the new marker, INSM1, can be expressed in thymic carcinomas [9,30].

Thus, we further investigated the expression of genes related to neuroendocrine lineages (which refers to tuft cells, ionocytes, and neuroendocrine cells in this study) in normal TECs and thymic carcinomas. This investigation utilized several independent datasets and our own cohort, with the aim of enhancing our understanding of the tumor and identifying new therapeutic targets. Additionally, we explored which signatures were lost in thymic carcinomas and, finally, discussed the relevance of DNA methylation for the unique expression profiles.

## 2. Materials and Methods

### 2.1. Analysis of a Publicly Available Single-Cell RNA Sequencing Dataset of the Normal Human Thymus

We analyzed a single-cell RNA sequencing (scRNA-seq) dataset of the normal human thymus (GSE147520) [22]. This dataset comprised two fetal (19 and 23 weeks), two postnatal (both 10 months), and one adult (25 years old) thymus. We initially focused on the postnatal and adult thymuses, as TETs do not occur in fetuses [10]. The scRNA-seq data was read using Python (ver. 3.8.16) and Scanpy (ver. 1.9.3). We excluded genes expressed in less than three cells, cells expressing < 200 genes or >5000 genes, and cells where the counts of mitochondrial genes accounted for >10% of all counts, as these data were deemed unreliable. We first concatenated three datasets and extracted the epithelial cells based on the original author’s annotation [22]. After log-normalizing and scaling counts, we computed the PCA and created a neighborhood graph. Batch effect correction was conducted using BBKNN [31], and clustering was performed using the Leiden algorithm. This resulted in the re-assortment of 3825 epithelial cells into 16 clusters, visualized using UMAP (Figure 1a). To analyze the. genes expressed in a small number of cells, we employed the noise reduction method, RECODE [32].

We observed that the distribution of TECs significantly differed between the postnatal and adult thymuses (Figure 1b). The number and distribution of *POU2F3*, the tuft cell master regulator [33], also varied (Figure 1c). In addition to the abovementioned differences, given that TETs are predominantly adult tumors [10], we decided to exclude TECs from the postnatal thymus from our analysis (Figure 1d). Each gene of interest was mapped on the final UMAP with only adult TECs. Each cluster was categorized into one of eight groups [cTECs, immature TEC, mTEC lo (low), mTEC hi (high), corneocyte-like, neuroendocrine, myoid, and myelin +] as per the original article (Figure 1e), based on the expression patterns of the known marker genes [22] (Appendix A). The neuroendocrine group, which is our study focus, comprised clusters 9 and 15 in our analysis. To investigate the gene expression patterns within these two clusters, we subset them and computed the PCA, a neighborhood graph, and UMAP again.

### 2.2. Analysis of Publicly Available Bulk mRNA Expression and Methylation Datasets of Thymic Epithelial Tumors and Lung Cancers

We conducted an analysis on four bulk RNA-seq datasets of TETs and lung cancers, which are as follows: (1) a thymoma dataset from the TCGA PanCancer Atlas, which includes thymic carcinomas and thymic neuroendocrine neoplasms (NENs) (cBioPortal, http://www.cbioportal.org (accessed on 10 March 2023)) [17,18]. This dataset (N = 117) comprised various TET subtypes, including type A (N = 10), AB (N = 48), B1 (N = 12), B2 (N = 25), and B3 (N = 10) thymomas, micronodular thymoma (N = 2), thymic carcinoma (N = 9), and thymic NEN (N = 1). For this study, we excluded micronodular thymoma and thymic NEN due to their small numbers for statistical analysis; (2) an RNA-seq dataset of TETs provided by Petrini et al. (GSE57892) [34]. This dataset (N = 22) comprised various TET subtypes, including type A (N = 5), AB (N = 2), B2 (N = 3), and B3 thymomas (N = 5), and thymic carcinoma (N = 7). In this cohort, we combined type A and AB thymomas into one group and type B2 and B3 thymomas into another, with consideration of their biological similarities [10,34], and for subsequent statistical analyses; (3) an RNA-seq dataset of 81 SCLCs from cBioPortal [35]. These SCLCs were categorized into 11 tuft cell-like and 70 non-tuft cell-like subsets, as previously mentioned [36]; and (4) an RNA-seq dataset comprising 66 large cell neuroendocrine carcinomas (LCNEC) of the lung [37]. Similar to previous work, these LCNECs were divided into 12 tuft cell-like and 44 non-tuft cell-like subsets [19]. Additionally, we analyzed the DNA methylation dataset of the thymomas from TCGA, Firehose Legacy (cBioPortal). The cohort (N = 117) matched the ones from Thymoma, TCGA PanCancer Atlas.

### 2.3. Analysis of Our Cohort and Real-Time Quantitative Polymerase Chain Reaction

We assessed the mRNA expression levels by performing a real-time quantitative PCR (qPCR) in our study cohort, comprising 27 snap-frozen cases of epithelium-rich TETs, including type A (N = 7) and B3 (N = 8) thymomas, as well as thymic carcinomas (N = 12). These specimens were obtained from the Institute of Pathology, University Medical Center Mannheim, and the Medical Faculty Mannheim, Heidelberg University. Total RNA was extracted using TRIzol (Thermo Fisher Scientific, Waltham, MA, USA), and 500 ng of each RNA was reverse-transcribed using the PrimeScript RT reagent KIT (Takara, Shiga, Japan). The qPCR analysis was conducted using SYBR Premix Ex Taq (Takara) and analyzed with a Step-One Real-Time PCR System (Thermo Fisher Scientific). *Beta-2 microglobulin (B2M)* served as the housekeeping gene, and the tonsil’s expression level was set as the reference. The primer pairs used for the experiments were as follows: *NFATC1*: forward 5′-ttcgggagaggagaaactttgg-3′, reverse 5′-tggaggatgcatagccatagtg-3′. *B2M*: forward 5′-actctctctttctggcctgg, reverse 5′-gacaagtctgaatgctccact-3′.

### 2.4. Statistical Analysis

The differences in the continuous variables used in the RNA-seq and qPCR analyses were evaluated using the Wilcoxon test. The correlation between the mRNA expression level (RNA-seq) and methylation status of the genes of interest was evaluated by Pearson’s correlation. The differences at a *p*-value < 0.05 were considered significant. All the statistical analyses were performed with JMP16 (Statistical Analysis System, Cary, NC, USA).

## 3. Results

### 3.1. The Presence of Tuft Cells and Ionocytes within the Neuroendocrine Group in the Adult Thymus

As detailed in the Materials and Methods section, our analysis focused specifically on TECs from the adult thymus. Consequently, the number of evaluated TECs was smaller compared to that of the original study, which included TECs from fetal, postnatal, and adult thymuses [22]. Nevertheless, within the neuroendocrine group, we were able to identify cells co-expressing representative tuft cell genes (*POU2F3*, *GFI1B*, *TRPM5*, *CHAT*) [19,29] in a limited area (Figure 2a). The co-factors of *POU2F3*, specifically *POU2AF2/C11ORF53* and *POU2AF3/COLCA2* [38,39,40], were also expressed in these cells, although the *POU2AF3* expression extended over the neuroendocrine group (Figure 2a). We had already demonstrated that these tuft cell genes were significantly upregulated in thymic carcinomas compared to thymomas [19,20]. Similarly, the *POU2F3* co-factors showed significant expressions in thymic carcinomas compared to thymomas in both the TCGA [16] and Petrini et al. datasets [34] (Figure 2b,c).

Our prior research demonstrated that tuft cell-like cancers also significantly express markers of ionocytes, another distinct type of epithelial cells [20,23,24,36]. In line with the original study [22], we identified cells co-expressing *FOXI1* (master regulator of ionocytes) and *CFTR* (encoding the most representative functional molecule) [23,24] within the adult thymus. These cells formed a distinct population within the neuroendocrine cluster, roughly corresponding to cluster 15 (Figure 2d). A significant expression of *FOXI1* and *CFTR* in thymic carcinomas, compared to thymomas, was evident in the aforementioned two datasets [16,34], except for *CFTR* in the TCGA dataset (Figure 2c,e). Our findings indicate the presence of tuft cells and ionocytes in the adult thymus, where virtually all TETs develop [10]. Furthermore, thymic carcinomas exhibited a significant expression of key tuft cell- and ionocyte-related genes compared to thymomas.

### 3.2. Expression of General Neuroendocrine Markers, except for INSM1, Is Not Remarkable in Thymic Carcinomas

Subsequently, we analyzed the expression of more general neuroendocrine markers within normal adult TECs and TETs. TECs expressing *BEX1*, *NEUROD1*, *CHGA*, *SYP*, and *INSM1* were notably enriched in the neuroendocrine group, primarily in the upper region (Figure 3a). Surprisingly, *ASCL1* expression was observed in most TECs across the lineages in our analysis (Figure 3a). When we examined the expression of these genes in TETs, we found that *INSM1* showed a significant expression in thymic carcinomas compared to thymomas, which aligns with the findings of a previous study [9]. However, the expression of the other markers did not exhibit pronounced differences (Figure 3b,c). This observation seems consistent with the consensus that classical neuroendocrine markers, including chromogranin A (CHGA) and synaptophysin (SYP), may be expressed in thymic carcinomas but are generally focal in immunohistochemistry [10].

In an effort to understand why markers of tuft cells and ionocytes are preferentially expressed in thymic carcinomas, we investigated the distribution of the common squamous cell markers, *KRT5* and *TP63* [10], in the adult thymus. This is relevant because bona fide NENs lack the expression of squamous cell markers, but they are typically expressed in thymic SQCCs [10]. Interestingly, while most TECs expressed these squamous cell markers, cells within the neuroendocrine group that strongly expressed general neuroendocrine markers exhibited a weak expression of *KRT5* and rarely expressed *TP63*. Contrarily, tuft cells and ionocytes, especially the latter, seemed to maintain their expression (Figure 3d). These distinctions became more evident when we focused solely on the neuroendocrine group with a higher resolution (Figure 4).

### 3.3. Genes Related to Thymic Tuft Cell Development Are Biasedly Expressed in the Neuroendocrine Group in the Adult Thymus and Significantly Expressed in Thymic Carcinomas

Subsequently, we explored the mechanisms behind the gene expression signatures unique to thymic carcinomas. Although significant progress has been made in understanding the genes that differentiate and proliferate thymic tuft cells, the developmental process of ionocytes and neuroendocrine cells in the thymus has remained relatively unexplored. Therefore, we conducted a literature search using the keywords “tuft cells AND thymus” and drew insights from discussions at the recent international conference on tuft cells (Tuft Cells 2023). We identified the following nine genes for investigation: *IL33*, *DHX9*, *LTBR*, *SOX4*, *HIPK2*, *IL4R*, *SIRT6*, *HDAC9*, and *FEZF2* [11,41,42,43,44]. These genes were then analyzed for their expression in adult TECs and TETs. Among these genes, *HIPK2*, *SIRT6*, *HDAC9*, and *FEZF2* exhibited substantial expression, primarily within the neuroendocrine group (Figure 5a). Furthermore, *HIPK2*, *HDAC9*, and *FEZF2* displayed a significantly higher expression in thymic carcinomas than in thymomas in both datasets (Figure 5b,c, and Appendix A). Notably, among the HDAC family, only *HDAC9* showed a significant expression in thymic carcinomas compared to thymomas (Figure 5b and Appendix A).

When we included SCLC and LCNEC in the lung in the examination, no remarkable trends were observed between the tuft cell-like and non-tuft cell-like subsets for the expression of *HIPK2*, *HDAC9*, and *FEZF2* (Figure 5d,e). These findings suggest that the acquisition of tuft cell-like properties in TETs is intricately related to the normal developmental process of thymic tuft cells.

### 3.4. NFATC1 Expression in Thymic Carcinoma and Ionocytes in the Adult Thymus

The most classical and reliable markers for thymic SQCC are KIT and CD5 [10,25,26,27,45]. The strong correlation between KIT expression and the tuft cell lineage [19,20,29], along with the observation that *CD5* expression was relatively confined to the neuroendocrine group [as previously described [7] and confirmed even in adult TECs alone (shown later)], suggested that CD5 expression might be associated with specific transcription factors related to neuroendocrine lineages.

CD5 is traditionally known as a surface molecule of lymphocytes, and its regulatory mechanism has been well-studied in this cell type. Through a literature search, we identified the ETS family, NFATC1, and STAT3 as possible positive regulators [46,47,48,49]. When we assessed their expression in TETs, *NFATC1* was significantly expressed in the thymic carcinomas in the TCGA dataset (Figure 6a), whereas the expression of the other genes did not show remarkable differences (not shown). Higher *NFATC1* expression was also observed in Petrini et al.’s [34] dataset and in the qPCR analyses with TETs from our tissue samples, although it may not have reached statistical significance (Figure 6b,c). Furthermore, we examined the normal thymus and discovered that *NFATC1* was highly expressed in the neuroendocrine groups, particularly in ionocytes, although this expression did not appear to completely overlap with *CD5* (Figure 6d).

### 3.5. Expression of Pan-Thymic Epithelial Markers Is Suppressed in Thymic Carcinoma

Next, we focused on genes that are suppressed in thymic carcinomas. The absence of thymus-like cytoarchitecture in thymic carcinomas [10] suggests that this lack of thymus-like features may also manifest at the gene expression level. Therefore, we examined the expression of pan-thymic epithelial markers, including *PAX9*, *SIX1*, and *FOXN1*, which are known to play essential roles in thymic organogenesis [22,50,51,52,53]. In our analysis, most TECs in the adult thymus expressed *PAX9* and *SIX1* (Figure 7a). Notably, the expression of these markers was significantly suppressed in thymic carcinomas compared to thymomas (Figure 7b,c). Other genes related to thymic organogenesis, including *TBX1*, *PAX1*, *HOXA3*, and *EYA1* [51], exhibited similar trends (Appendix A). In our analysis, *FOXN1* expression was not universally distributed but rather substantially limited in adult TECs, and its expression did not show remarkable differences among TETs (Figure 7a–c).

### 3.6. Characteristic DNA Methylation Patterns May Be Linked to the Expression Profiles of Thymic Carcinoma

Finally, we sought to understand how the unique expression profiles of thymic carcinomas are acquired. The TCGA data have previously shown that global methylation patterns are highly correlated with the histological subtypes of TETs [16]. Therefore, we investigated the correlation between methylation and mRNA expression status in the genes newly identified as activated or suppressed in thymic carcinomas, specifically *HIPK2*, *HDAC9*, *NFATC1*, *PAX9*, and *SIX1* (the methylation status of *FEZF2* was not available in the dataset). All of these genes exhibited negative correlations between the proportion of methylation and mRNA expression, particularly those associated with pan-thymic epithelial markers (Figure 7d). This suggests the relevance of epigenetic regulation in shaping the properties of thymic carcinoma.

## 4. Discussion

Focusing on the so-called neuroendocrine phenotype, we explored the under-recognized expression profiles of TETs, especially thymic carcinomas. We discovered that: (1) thymic carcinomas overexpressed *HDAC9* and *NFATC1*, which could be potential therapeutic targets and associated with the neuroendocrine phenotype in the thymus; (2) genes involved in thymic development, including *PAX9* and *SIX1*, were markedly downregulated in thymic carcinomas; and (3) thymic carcinoma had distinct gene expression patterns that might be related to the global methylation changes.

Zhang et al. [44] proposed that *Hdac9* is important for thymic tuft cell development, according to a mouse study. They reported that *Sirt6* deletion increased *Hdac9* and decreased *Pou2f3*-regulated gene expression without altering the *Pou2f3* expression. They also noted that *Sirt6* deletion activated the “NF-kappa B signaling pathway” and “pathways in cancer”. Thus, we hypothesized that Hdac9/HDAC9 might inhibit the terminal differentiation of Pou2f3/POU2F3-positive tuft cell precursors and participate in their oncogenic behaviors. This seems consistent with the observation that tuft cell-like carcinomas, including thymic carcinoma, are not fully differentiated [20]. Moreover, the selective upregulation of *HDAC9* in thymic carcinomas compared to thymomas implies that thymic carcinomas may strongly rely on HDAC9 for survival. HDAC inhibitors have been explored as anticancer agents [54], and some are clinically approved [55]. Therefore, the HDAC inhibitors targeting HDAC9 could be a promising treatment option for inoperable thymic carcinomas if the relevant role of HDAC9 in TECs and TETs, especially in the neuroendocrine lineage, is clarified.

When exploring the possible regulators of CD5 expression in thymic carcinomas, we identified *NFATC1* upregulation in the tumor. Although it cannot be concluded that NFATC1 is the upstream regulator of CD5, NFAT proteins have been proposed as potential therapeutic targets and should be considered when deciding on treatment strategies [55,56,57]. Given that there is limited existing research on NFATC1 in the context of neuroendocrine lineages, particularly ionocytes, this can be an avenue for future investigation. Furthermore, the relationship between CD5 and neuroendocrine lineages across different organs is of interest, as Merkel cell carcinoma, a cutaneous neuroendocrine carcinoma, can express this molecule [58].

In contrast to the unique neuroendocrine marker expression, there appeared to be a relatively negative correlation between the general neuroendocrine markers and squamous cell markers in thymic carcinomas. Additionally, the suppression of general thymic epithelium markers, represented by *PAX9* and *SIX1*, was observed in thymic carcinomas. This finding seems consistent with the lack of morphological thymus-like features in this tumor. Additionally, the limited expression of *FOXN1*, a well-known gene expressed in TECs [56], underscores the substantial differences in the cellular composition between the fetal/postnatal and adult thymuses.

Lastly, we demonstrated that DNA methylation plays a substantial role in regulating the unique expression profiles of thymic carcinomas. Thus, it is essential to understand how these characteristic methylation patterns are acquired. We speculate that the loss of 16q, the most common chromosomal abnormality in thymic carcinomas [10,16], might be linked to the cancer-specific methylation patterns. Indeed, chromosomal abnormalities can lead to changes in genome-wide gene expression, with DNA methylation playing a role in this process [59].

The present study has several limitations that should be acknowledged. First, most of the results were derived from the analysis of publicly available datasets on mRNA and DNA methylation, and they were not validated using our own research samples. This reliance on existing datasets, although reliable, introduces a potential source of bias. Additionally, the lack of protein expression data is a substantial drawback in the study, especially when discussing drug targets. Second, the scRNA-seq data of the adult thymus was based on the analysis of only one thymus, and the patient’s age was relatively young (25 years) considering that TETs typically occur in middle- to old-aged patients [10]. Given that the cellular constituents varied even between the two postnatal thymus samples analyzed, it is reasonable to assume adult thymuses slightly differ in terms of their TEC populations. Obtaining sufficient epithelial cells from non-neoplastic, old-aged thymus samples may be challenging due to age-related atrophy. However, if the methods for efficiently extracting TECs from old-aged thymus samples can be established, they would contribute to gaining a deeper understanding of the factors contributing to TET occurrence. Third, the results of this study are observational, and no definitive biological significance has been drawn. As a result, further in vitro and in vivo studies are necessary to validate our findings, particularly those related to the activation of *HDAC9* and *NFATC1* in thymic carcinomas, before they can be applied in a clinical context. Despite these limitations, we believe that this study holds significance in advancing our understanding of the biology of thymic carcinoma, especially with regard to the neuroendocrine phenotype.

## 5. Conclusions

The unique expression patterns of thymic carcinomas can be partially attributed to their neuroendocrine characteristics, particularly their resemblance to specific neuroendocrine cells, such as tuft cells and ionocytes. *HDAC9* and *NFATC1*, which were characteristically expressed in the neuroendocrine group in adult TECs and thymic carcinomas, should be investigated further due to their potential biological significance in TECs and their possibility of being therapeutic targets of thymic carcinomas. The downregulation of the markers common to all TECs is another feature of the gene expression profiles of thymic carcinomas. DNA methylation status may be associated with these newly discovered signatures of thymic carcinomas.

## Figures and Tables

**Figure 1 cancers-16-00115-f001:**
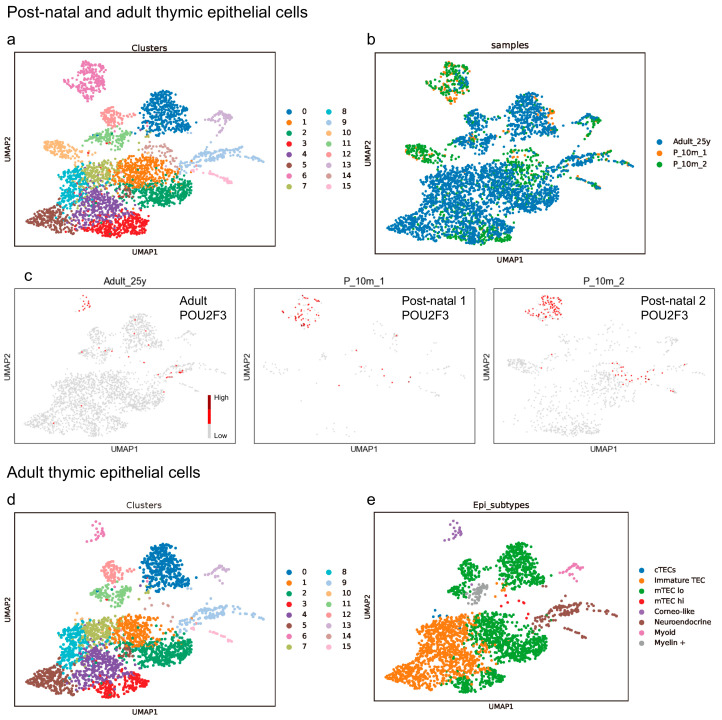
Profiling of human thymic epithelial cells. (**a**) Re-analysis of publicly available single-cell RNA sequencing (scRNA-seq) data reveals that the adult (N = 1, 25 years old) and postnatal (N = 2, both 10 months) thymic epithelial cells (TECs) can be categorized into 16 clusters. (**b**,**c**) However, the distribution of TECs (**b**), including those expressing POU2F3 (**c**), significantly differs between the postnatal and adult thymuses. (**d**) Adult TECs alone also can be separated into 16 clusters. (**e**) Each cluster was assigned to one of eight cell lineages, including cortical TECs (cTECs), immature TECs, medullary TECs, mTEC lo (low), mTEC hi (high), corneocyte-like, neuroendocrine, myoid, and myelin +, as originally described, based on the expression patterns of the marker genes [22]. The neuroendocrine group encompasses clusters 9 and 15 ((**a**–**e**) Bautista et al., 2021 [22]. GSE1475220).

**Figure 2 cancers-16-00115-f002:**
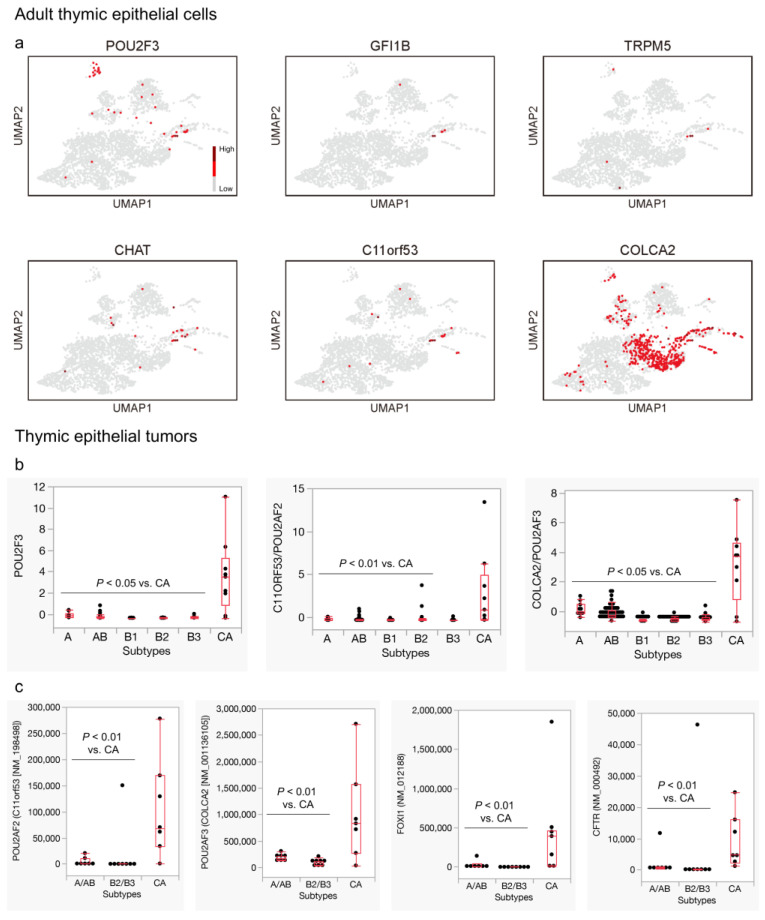
Expression of representative tuft cell- and ionocyte-related genes in adult thymic epithelial cells (TECs) and thymic epithelial tumors. (**a**) A subset of adult TECs expresses representative tuft cell markers (*POU2F3*, *GFI1B*, *TRPM5*, *CHAT*) and co-factors of *POU2F3* (*POU2AF2/C11orf53* and *POU2AF3/COLCA2*). TECs expressing these genes are predominantly found in a confined area, which is closely associated with cluster 9, especially adjacent to cluster 15. (**b**) Thymic carcinomas significantly upregulate *POU2F3*, *POU2AF3*, and *POU2AF3* compared to thymomas (*p* < 0.05), except for *POU2AF3* when compared to type B3 thymoma (*p* = 0.07, Wilcoxon test). (**c**) *POU2AF3*, *POU2AF3*, and representative ionocyte markers (*FOXI1* and *CFTR*) are significantly upregulated in thymic carcinomas compared to thymomas in another dataset (*p* < 0.05, Wilcoxon test). (**d**) A subset of adult TECs expresses *FOXI1* and *CFTR* and is enriched in cluster 9, partly overlapping with cells expressing most tuft cell markers. (**e**) Thymic carcinomas significantly upregulate *FOXI1* compared to thymomas. Regarding *CFTR*, the thymic carcinoma shows almost a significant upregulation of this gene (*p* = 0.06, Wilcoxon test) when compared to type B1 thymoma, but not when compared to other thymoma subtypes ((**a**,**d**) Bautista et al., 2021 [22]. GSE1475220; (**b**,**e**) Thymoma, TCGA PanCancer Atlas [Y-axis: RNA-seq, mRNA Z score]; (**c**) Petrini et al., 2014 [34] [Y-axis: RNA-seq, fragments per kilobase of exon per million reads mapped, FPKM]).

**Figure 3 cancers-16-00115-f003:**
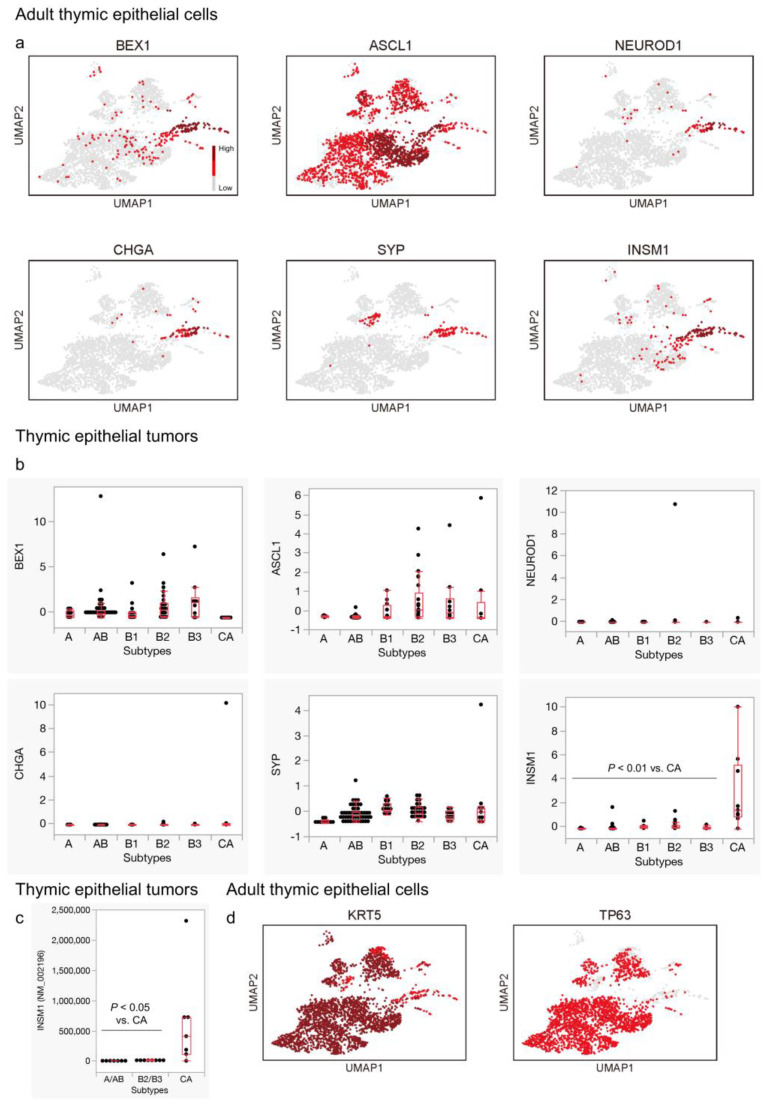
Expression of general neuroendocrine markers in adult thymic epithelial cells and thymic epithelial tumors. (**a**) General neuroendocrine markers, including *BEX1*, *NEUROD1*, *CHGA*, *SYP*, and *INSM1*, exhibit characteristic expression patterns in the neuroendocrine group among the adult thymuses. Surprisingly, *ASCL1* shows broad expression across different cell types. (**b**) *INSM1* displays significant upregulation in thymic carcinomas compared to thymomas (*p* < 0.01, Wilcoxon test), whereas the differences in expression of the other genes between thymic carcinomas and thymomas are less pronounced. (**c**) Higher expression of *INSM1* in thymic carcinomas than in thymomas is consistent with the findings from another dataset (*p* < 0.05, Wilcoxon test). (**d**) The common squamous cell markers, *KRT5* and *TP63*, are expressed in many adult thymic epithelial cells (TECs). Notably, TECs with a strong expression of general neuroendocrine markers (enriched in the upper half of cluster 9) tend to exhibit weaker *KRT5* expression compared to other cells and do not express *TP63* ((**a**,**c**) Bautista et al., 2021 [22]. GSE1475220; (**b**) Thymoma, TCGA PanCancer Atlas [Y-axis: RNA-seq, mRNA Z score]).

**Figure 4 cancers-16-00115-f004:**
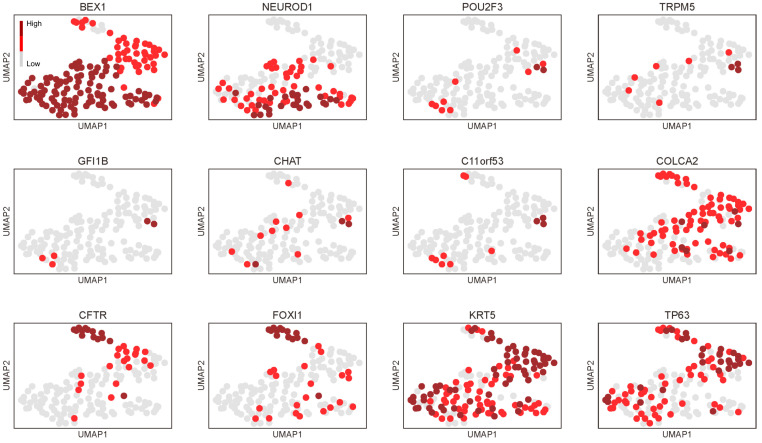
Profiling of adult thymic epithelial cells focusing on the neuroendocrine group with a higher resolution. Neuroendocrine cells (high in *BEX1/NEUROD1*), tuft cells (high in *POU2F3* to *C11ORF53/POU2AF2*, and *COLCA2/POU2AF3*), and ionocytes (high in *FOXI1/CFTR*) can be distinctly identified. Neuroendocrine cells tend to exhibit a weaker expression of squamous cell markers (*KRT5/TP63*), whereas ionocytes retain their expression.

**Figure 5 cancers-16-00115-f005:**
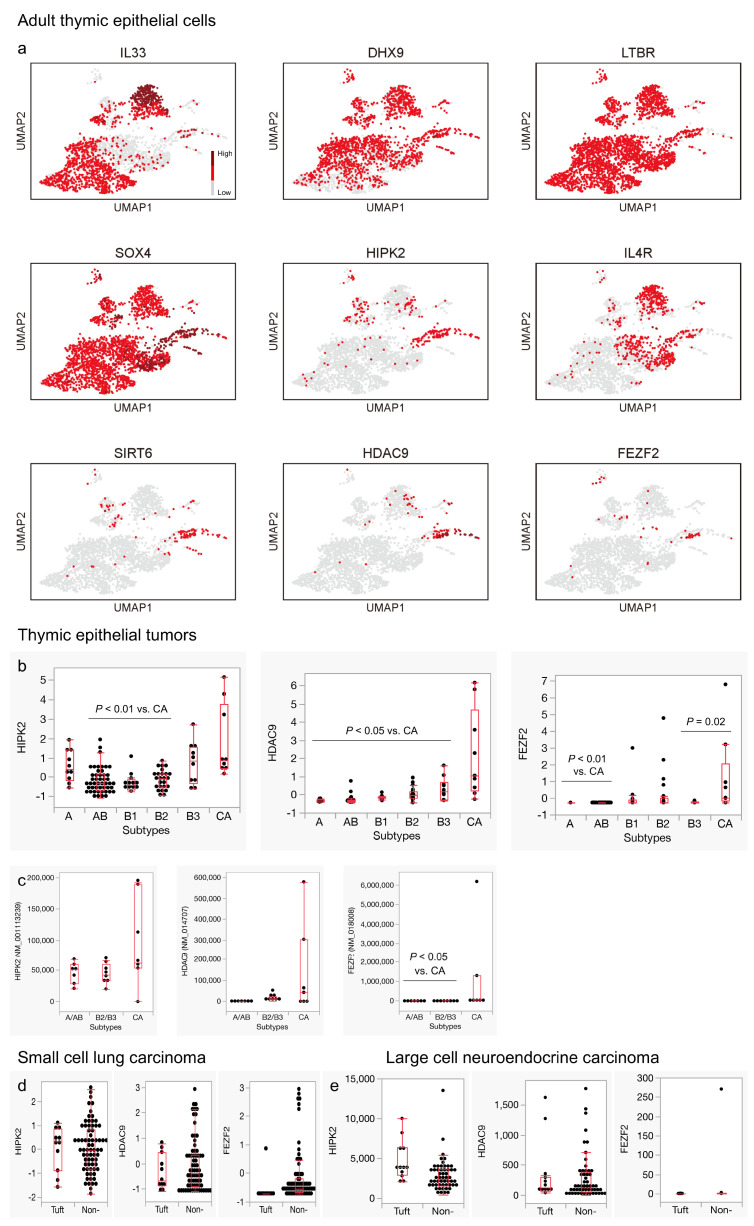
Expression of genes potentially regulating tuft cell development in adult thymic epithelial cells and thymic epithelial tumors. (**a**) Through a literature search, nine genes were identified as potential regulators of thymic tuft cell development: *IL33*, *DHX9*, *LTBR*, *SOX4*, *HIPK2*, *IL4R*, *SIRT6*, *HDAC9*, and *FEZF2*. *HIPK2*, *SIRT6*, *HDAC9*, and *FEZF2* exhibit relatively selective expressions within the neuroendocrine group. (**b**) *HIPK2*, *HDAC9*, and *FEZF2* are significantly upregulated in thymic carcinomas (*HIPK2*: *p* = 0.18 for type A vs. thymic carcinoma (CA); *p* < 0.05 for AB, B1, B2 vs. CA; *p* = 0.39 for B3 vs. CA; *HDAC9*: *p* < 0.05 for all thymoma subtypes vs. CA; *FEZF2*: *p* < 0.05 for type A, AB, B3 vs. CA; *p* = 0.08 for B1 vs. CA; *p* = 0.09 for B2 vs. CA, Wilcoxon test). (**c**) A higher expression of *HIPK2*, *HDAC9*, and *FEZF2* was also observed in another dataset (*HIPK2*: *p* = 0.13 for type A/AB vs. CA; *p* = 0.12 for B2/B3 vs. CA; HDAC9: *p* = 0.25 for A/AB vs. CA; *p* = 0.77 for B2/B3 vs. CA; *FEZF2*: *p* < 0.05 for A/AB, B2/B3 vs. CA, Wilcoxon test). (**d**,**e**) The tuft cell-like subset does not exhibit a clear increase in *HIPK2*, *HDAC9*, and *FEZF2* expressions compared to the non-tuft cell-like subset in small cell carcinomas (**d**) and large cell neuroendocrine carcinomas (**e**) of the lung ((**a**) Bautista et al., 2021 [22]. GSE1475220; (**b**) Thymoma, TCGA PanCancer Atlas [Y-axis: RNA-seq, Z score]; (**c**) Petrini et al., 2014 [34] [Y-axis: RNA-seq, FPKM]; (**d**) George et al., 2015 [35] [Y-axis: RNA-seq Z score]; (**e**) George et al., 2018 [37] [Y-axis: RNA-seq, FPKM]).

**Figure 6 cancers-16-00115-f006:**
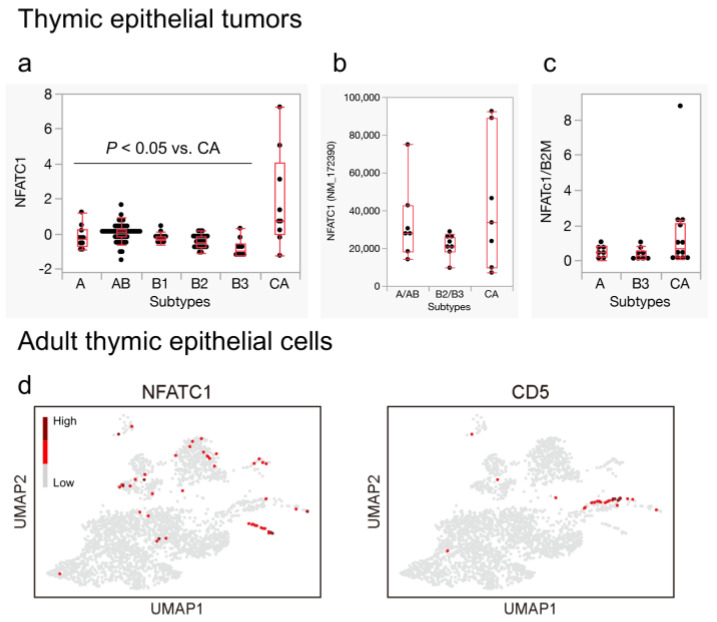
*NFATC1* expression in thymic epithelial tumors and adult thymic epithelial cells. (**a**,**b**) *NFATC1* exhibits significant expression in thymic carcinomas compared to thymomas in the TCGA dataset (*p* < 0.05, Wilcoxon test). (**b**,**c**) Similar trends are observed in another publicly available RNA-seq dataset and our quantitative real-time PCR (qPCR) with our own samples, although statistical significance is not reached. (**d**) *NFATC1* is characteristically expressed in cluster 15, which is nearly identical to ionocytes. *CD5*-expressing cells are found in a distinct area within the neuroendocrine group ((**a**) Thymoma, TCGA PanCancer Atlas [Y-axis: RNA-seq, Z score]; (**b**) Petrini et al., 2014 [34] [Y-axis: RNA-seq, FPKM]; (**c**) qPCR with our sample; (**d**) Bautista et al., 2021 [22]. GSE1475220).

**Figure 7 cancers-16-00115-f007:**
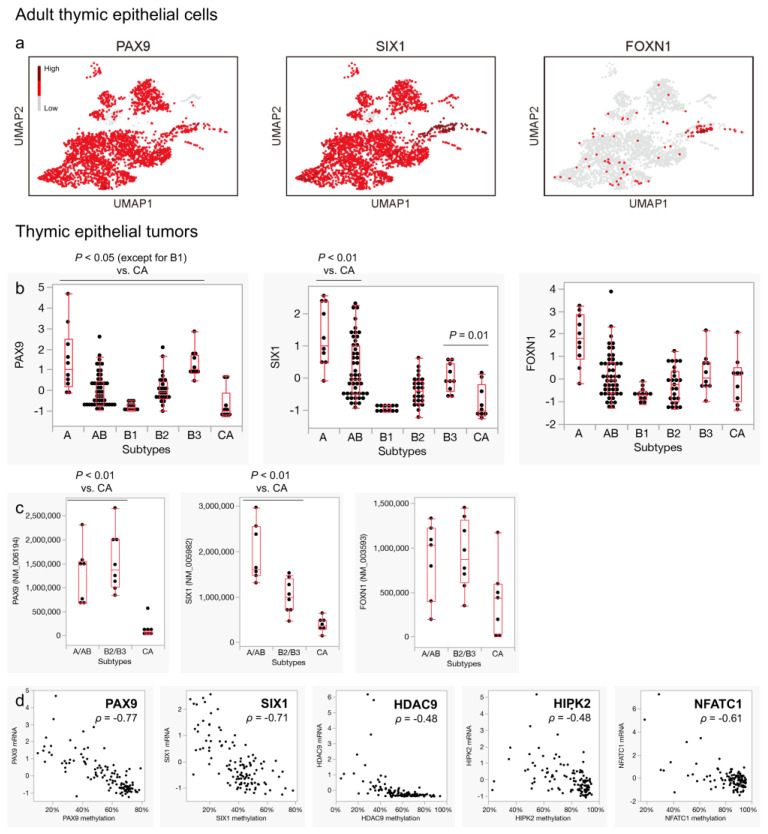
Expression of pan-thymic epithelium markers in thymic epithelial cells/tumors and the correlation with mRNA expression and methylation status of the genes of interest. (**a**) *PAX9* and *SIX1*, which are general pan-thymic epithelium markers, are widely expressed in adult thymic epithelial cells (TECs), whereas *FOXN1* expression is relatively limited. (**b**) The expressions of *PAX9* and *SIX1* are significantly suppressed in thymic carcinomas compared to thymomas (*PAX9*: *p* < 0.05 for type A, AB, B2, B3 vs. thymic carcinoma [CA], *p* = 0.17 for B1 vs. CA; *SIX1*: *p* < 0.05 for type A, AB, B3 vs. CA, *p* = 0.64 for B1 vs. CA, *p* = 0.05 for B2 vs. CA). The lower expression level in type B1 thymoma is likely because the majority of RNAs originated not from neoplastic epithelial cells but from the accompanying immature T cells. A significantly lower expression of *FOXN1* in thymic carcinoma is not evident. (**c**) A lower expression of *PAX9* and *SIX1* in thymic carcinomas than in thymomas is consistent with the findings obtained from another dataset (*p* < 0.05, Wilcoxon test). (**d**) mRNA expression of genes that are significantly downregulated (*PAX9* and *SIX1*) and upregulated (*HDAC9*, *HIPK2*, and *NFATC1*) is negatively correlated with their methylation proportion ((**a**) Bautista et al., 2021 [22] GSE1475220; (**b**) Thymoma, TCGA, PanCancer Atlas [Y-axis: RNA-seq, Z score]; (**c**) Petrini et al., 2014 [34] [Y-axis: RNA-seq, FPKM]; (**d**) Thymoma, TCGA, PanCancer Atlas and Thymoma, Firehose [X-axis: methylation proportion, Y-axis: RNA-seq, Z score, ρ values indicate Pearson’s correlation]).

## Data Availability

The datasets used and/or analyzed during the current study are available from the corresponding author on reasonable request.

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
