# Peer review of "Thymic Carcinoma: Unraveling Neuroendocrine Differentiation and Epithelial Cell Identity Loss"

_cancers, 2023, doi:10.3390/cancers16010115_

Round 1

Reviewer 1 Report

Comments and Suggestions for Authors

In this study, the signs of thymus cancer gene expression were investigated in tuft cells and their relatives - ionocytes and neuroendocrine cells (neuroendocrine group).
In this study, publicly available datasets on mRNA expression and methylation status of TECs and lung cancers were examined. Quantitative real-time PCR was also performed with tissue samples. Thyme carcinoma showed a neuroendocrine phenotype towards tuft cells and oenocytes. By investigating the possible regulators of this phenotype, it was found that HDAC9 and NFATC1 are specifically expressed in the neuroendocrine group in adult TECs and thymic carcinoma. In addition, panthymic epithelial markers, exemplified by PAX9 and SIX1, were significantly suppressed in thymic carcinoma. Thymic carcinoma can be characterized by its unique neuroendocrine differentiation and loss of identity as thymic epithelial cells. HDAC9 and NFATC1 may provide new targets for the treatment of this apparent malignancy.

Reviewer 2 Report

Comments and Suggestions for Authors

Overall work is good and considerable for publication after minor corrections which are as follows,

1. Remove the grammatical mistakes.

2. Check the English language.

3. Conclusion is very short which should be refine and increase with relevant results. 

Comments on the Quality of English Language

 Minor editing of English language required.

Reviewer 3 Report

Comments and Suggestions for Authors

The manuscript entitled "Thymic Carcinoma: Unraveling Neuroendocrine Differentiation and Epithelial Cell Identity Loss" Yamada et al is well written and addresses an important issue providing novel insights into pathophysiology of thymic carcinoma even despite some limitations correctly acknowledged by the authors.

Minor issues:

-          Provide developer and developer’s country for JMP16 (software used for statistical analysis)

The study focuses on analysis of neuroendocrine phenotypes in thymic carcinoma with identification of genes with dysregulated expression to explore their possible therapeutic targeting.

The topic is of interest and is worth being addressed.

The study identified abnormal expression patterns of HDAC9 and NFATC1 suggesting that they can be used as potential targets for therapeutic use .

The authors used only publicly available tools to assess gene expression, which can be expanded. However, it was acknowledged as a limitation.

Conclusions are supported by the findings.

References are appropriate.

Figures and tables present the data well. 

Reviewer 4 Report

Comments and Suggestions for Authors

Congratulations on your great study! I think this study used a lot of references and a public database. 

I have one question about how you proved these genes are related to thymic malignancy. However, are there any experimental results that prove the relationship? If there is not, this result may be nonsensical.

Reviewer 5 Report

Comments and Suggestions for Authors

The authors examined the neuroendocrine phenotypes of thymic carcinoma and identified a few upregulated genes that are potentially associated with the phenotypes. Their findings revealed expression of HDAC9 and NFATC1 in the neuroendocrine group of adult thymic epithelial cells. This article should be reviewed again if the authors made some modifications.

  1. All of the pictures are low resolution.
  2. The legend of the picture should be carefully modified and checked. Some errors have occurred like Figure 1.
  3. Target genes like HDAC9and NFATC1 lack gene function analysis. Gene expression alone is not sufficient to support the authors' conclusions.
